# Risk Due to ABO Incompatibility and Donor-Recipient Weight Mismatch in Living Donor Kidney Transplantation: A National Cohort Study

**DOI:** 10.3390/jcm10235674

**Published:** 2021-12-01

**Authors:** Jun Young Lee, Sung Hwan Cha, Sung Hwa Kim, Kyung Hwan Jeong, Ku Yong Chung, Hong Rae Cho, Juhan Lee, Kyu Ha Huh, Jaeseok Yang, Myoung Soo Kim, Deok Gie Kim

**Affiliations:** 1Transplantation Center, Wonju Severance Christian Hospital, Wonju 26426, Korea; junyoung07@yonsei.ac.kr; 2Department of Nephrology, Yonsei University Wonju College of Medicine, Wonju 26426, Korea; 3Department of Surgery, Yonsei University Wonju College of Medicine, Wonju 26426, Korea; luvnya82@yonsei.ac.kr; 4Department of Biostatistics, Yonsei University Wonju College of Medicine, Wonju 26426, Korea; juniver1057@naver.com; 5Department of Internal Medicine, Kyung Hee University College of Medicine, Seoul 02447, Korea; khjeong@khu.ac.kr; 6Department of Surgery, Ewha Womans University Mokdong Hospital, Seoul 07985, Korea; kuyong@ewha.ac.kr; 7Department of Surgery, Ulsan University Hospital, Ulsan 44030, Korea; hrcho@uuh.ulsan.kr; 8Department of Surgery, Yonsei University College of Medicine, Seoul 03722, Korea; laplaine@yuhs.ac (J.L.); khhuh@yuhs.ac (K.H.H.); ysms91@yuhs.ac (M.S.K.); 9Department of Internal Medicine, Yonsei University College of Medicine, Seoul 03722, Korea; jcyjs@snu.ac.kr

**Keywords:** ABO incompatible, kidney transplantation, living donor, weight mismatch

## Abstract

The effect of donor-recipient weight mismatch is not well established in ABO-incompatible living donor kidney transplantation (LDKT). A total of 2584 LDKT patients in the Korean Organ Transplantation Registry were classified into four groups according to the presence or absence of ABO incompatibility and donor-recipient weight mismatch (donor-to-recipient weight ratio (DRWR) < 0.8). In a multivariable Cox analysis, the combination of ABO incompatibility and DRWR incompatibility (*n* = 124) was an independent risk factor for graft survival (HR = 2.73, 95% CI = 1.11–6.70) and patient survival (HR = 3.55, 95% CI = 1.39–9.04), whereas neither factor alone was a significant risk factor for either outcome. The combination of ABO incompatibility and DRWR incompatibility was not an independent risk factor for biopsy-proven graft rejection (HR = 1.27, 95% CI = 0.88–1.82); however, it was an independent risk factor for pneumonia (HR = 2.94, 95% CI = 1.64–5.57). The mortality rate due to infection was higher among patients with both ABO incompatibility and DRWR incompatibility than among patients with neither factor or with either factor alone. The combination of ABO incompatibility and DRWR incompatibility was an independent risk factor for graft and patient survival after LDKT, whereas neither factor alone significantly affected graft or patient survival. Thus, donor-recipient weight matching should be cautiously considered in LDKT with ABO incompatibility.

## 1. Introduction

Kidney transplantation (KT) is the best treatment for most patients with end-stage renal disease [1]. Amidst the circumstance of organ shortage, ABO-incompatible living donor KT (ABOi LDKT) is one way to expand the donor pool. Despite excellent outcomes of ABOi LDKT in an early study [2], there have been conflicting reports regarding the safety of ABOi LDKT compared to that of ABO-compatible living donor KT (ABOc LDKT) [3,4,5,6,7,8,9]. Furthermore, a recent meta-analysis found that ABOi LDKT resulted in lower short-term graft and patient survival compared with ABOc LDKT [10]. In contrast, a recent well-designed cohort study demonstrated that patients who underwent ABOi LDKT had improved survival compared with patients that waited for a transplant either from a deceased donor or an ABO-compatible live donor [11]. In this context, further risk analysis of ABOi LDKT in comparison with ABOc LDKT is warranted.

Nephron mass is an important determinant of the fate and function of the kidney [12], as insufficient nephron mass is thought to be the starting point of hyperfiltration, which ultimately leads to renal allograft damage [13,14]. Given that kidney size is associated with patient weight [15], several studies have emphasized the negative effect of low donor-to-recipient weight ratio (DRWR) on KT outcomes [16,17,18,19]. It is not known, however, if weight mismatch between donor and recipient interacts with ABOi when both risk factors are present. Therefore, we aimed to investigate whether the combination of ABOi and DRWR incompatibility (DRWRi) is an independent risk factor in living donor KT (LDKT).

## 2. Materials and Methods

### 2.1. Study Population

We conducted a retrospective cohort study of KT recipients in the Korean Organ Transplantation Registry (KOTRY) who underwent KT in Korea between May 2014 and December 2018. Details of the KOTRY are described elsewhere [20]. A total of 3050 LDKTs were in the KOTRY at the time of our study. We excluded cases with zero-human leukocyte antigen (HLA) mismatch (*n* = 209), crossmatch-positive LDKT (*n* = 252), or lack of data (*n* = 5). The remaining 2584 LDKTs included 1941 ABOc LDKTs and 643 ABOi LDKTs. We defined DRWRi as a DRWR less than 0.8, which was near the cutoff for the 25th percentile among all LDKTs in the dataset; we defined DRWR compatibility (DRWRc) as a DRWR greater than 0.8 (Appendix A). For our analysis, we divided the patients into four groups depending on the presence or absence of ABOc and DRWRc: ABOc-DRWRc (*n* = 1570), ABOc-DRWRi (*n* = 371), ABOi-DRWRc (*n* = 519), and ABOi-DRWRi (*n* = 124; Figure 1).

### 2.2. Data Collection

We retrieved data on the demographics of the recipients and donors, underlying kidney disease, donor-specific antibody (DSA) positivity, and induction agent. Serum creatinine was measured at the time of discharge, at 6 and 12 months, and annually thereafter. The estimated glomerular filtration rate (eGFR) was calculated from the serum creatinine level using the Modification of Diet in Renal Disease equation [21], which assigns a value of zero when graft failure occurs. Detailed data regarding ABOi LDKT, such as anti-blood group antibody and pretransplant desensitization, were unavailable for more than half of the sample, so we did not include them in our analysis (the available data are listed in Appendix A).

The causes of graft failure and patient death were recorded. We defined graft failure as a need for return to dialysis or re-transplantation. Data from patients who died with a functioning graft were censored in the analysis of graft failure. The first event of biopsy-proven rejection (BPAR) was recorded for each patient. The causative pathogens and infection sites were recorded for infections that required re-admission to the hospital after KT. The primary end point of the study was death-censored graft survival. Secondary end points were patient survival, BPAR, and infection.

### 2.3. Statistical Analysis

Categorical variables were compared by Chi-square test or Fisher’s exact test and presented as the number (proportion). Continuous variables were compared by one-way analysis of variance (ANOVA) and presented as the mean ± standard deviation. Survival curves were generated using the Kaplan–Meier method and compared by log-rank test. Cox regression analysis was performed for outcomes adjusted with covariates for which the *p*-value was <0.10 in a univariable Cox analysis. A multivariable model for patient survival was established by stepwise regression to ensure statistical significance with limited number of events. The results were presented as hazard ratios (HRs) with 95% confidence intervals (CIs). All analyses were performed using standard software (SPSS v23.0; IBM, Armonk, NY, USA, and R freeware v3.6.3, R Foundation for Statistical Computing, Vienna, Austria). *p* < 0.05 was considered statistically significant.

## 3. Results

### 3.1. Baseline Characteristics

The demographics of the study sample are shown in Table 1. The mean recipient age ranged from 47.1 to 49.8 years, with no significant difference among the four groups. Male recipients and female donors were predominant in the ABOc-DRWRi and ABOi-DRWRi groups, reflecting weight differences between the sexes. The body mass index (BMI) of the recipients was naturally higher in the two DRWRi groups than in the DRWRc groups, whereas that of the donors was vice versa. Genetically unrelated donors, kidney donation from a spouse, and HLA mismatch were all more common in the ABOi-DRWRi group than in the other three groups (Table 1). The mean duration of pre-transplant dialysis was similar among the four groups, ranging from 11.5 months to 17.8 months. Re-transplantation tended to be more frequent in the two DRWRc groups than in the DRWRi groups. Diabetes, hypertension, and cardiovascular disease were generally more frequent in the two DRWRi groups, reflecting the male predominance in these groups. The frequency of pre-transplant positivity for DSA with negativity for lymphocyte crossmatch (XM-DSA+) was 5.4% vs. 1.6% vs. 9.6% vs. 6.5% in the ABOc-DRWRc, ABOc-DRWRi, ABOi-DRWRc, and ABOi-DRWRi groups, respectively. Most patients used the IL-2 receptor antibody as an induction agent, although some in each group used anti-thymocyte globulin.

### 3.2. Graft Function

Figure 2 shows the mean eGFR among the four groups from the time of hospital discharge to 48 months after KT. At discharge, the mean eGFR was lower in the ABOc-DRWRi (59.0 mL/min/1.73 m^2^) and ABOi-DRWRi (57.8 mL/min/1.73 m^2^) groups than in the ABOc-DRWRc (75.3 mL/min/1.73 m^2^) and ABOi-DRWRc (75.2 mL/min/1.73 m^2^) groups. This trend was maintained throughout the study period. The full results are provided in Appendix A.

### 3.3. Graft Survival

During a mean follow-up of 36.2 ± 15.7 months, graft loss occurred in 50 (1.9%) patients. Rejection was the most common cause of graft loss (*n* = 23, 0.9%), the detailed causes of which are presented in Appendix A. In the Kaplan–Meier analysis, death-censored graft survival was lower in the ABOi-DRWRi group than in the other groups, although the difference was not statistically significant (*p* = 0.063, Figure 3a). The cumulative incidence of graft loss at 5 years was 2.7% vs. 3.0% vs. 2.3% vs. 5.0% in the ABOc-DRWRc, ABOc-DRWRi, ABOi-DRWRc, and ABOi-DRWRi groups, respectively. In univariable and multivariable Cox regression analyses (Table 2), ABOi and DRWRi were not independent risk factors for graft survival; however, the combination of both factors was (HR = 2.73, 95% CI = 1.11–6.70, *p* = 0.028), together with other factors such as age and the number of HLA mismatches. The full results of the Cox regression analyses of death-censored graft survival are provided in Appendix A.

### 3.4. Patient Survival

Thirty-one (1.2%) patients died during the study period. Infection was the most common cause of death (*n* = 16, 0.6%), followed by unknown causes (*n* = 7, 0.3%) and cardiovascular disease (*n* = 4, 0.2%). The detailed causes of death are presented in Appendix A. In the Kaplan–Meier analysis, patient survival was significantly lower in the ABOi-DRWRi group than in the other three groups (*p* < 0.001, Figure 3b). The cumulative mortality at 5 years was 1.4%, 0.8%, 1.2%, and 6.2% in the ABOc-DRWRc, ABOc-DRWRi, ABOi-DRWRc, and ABOi-DRWRi groups, respectively. In the Cox regression analysis (Table 2), ABOi and DRWRi were not independent risk factors for patient survival; however, the combination of both factors was (HR = 3.55, 95% CI = 1.39–9.04, *p* = 0.008), along with other factors such as cardiovascular disease, unrelated donor, and XM-DSA+ (Appendix A).

### 3.5. Biopsy-Proven Acute Rejection

In the Kaplan–Meier analysis, BPAR was more common in the ABOi-DRWRi group than in the other groups (*p* = 0.006, Figure 3c), although after adjustment with other covariates, ABOi-DRWRi (HR = 1.27, 95% CI = 0.88–1.82, *p* = 0.203), ABOi (HR = 1.14, 95% CI = 0.95–1.38, *p* = 0.190), and DRWRi (HR = 1.05, 95% CI = 0.84–1.31, *p* = 0.661) were not independent risk factors for BPAR (Table 2). The independent risk factors for BPAR were age, male sex, re-transplantation, donor age, number of HLA mismatches, and XM-DSA+ (Appendix A). For patients who did not experience BPAR, graft survival was similar among the four groups (*p* = 0.563, Figure 4a) Among patients who experienced BPAR, graft survival was lower in the ABOi-DRWRi group than in the other groups, although the difference was not statistically significant (*p* = 0.135, Figure 4b).

### 3.6. Infection

We compared the incidences of infections that required hospitalization within 1 year after KT (Table 3) and found no significant differences in the overall incidence of infection or the incidences of urinary tract infection, bacteremia, viral infection, and fungal infection among the four groups. In contrast, bacterial pneumonia, viral pneumonia, and Pneumocystis jiroveci pneumonia occurred more frequently in the ABOi-DRWRi group than in the other groups (Table 3). Kaplan–Meier analyses showed that cumulative incidence of pneumonia was significantly higher in the ABOi-DRWRi group than in the other groups (*p* < 0.001, Figure 3d). In univariable and multivariable Cox analyses of total pneumonia events during the study period, the combination of ABOi and DRWRi was an independent risk factor (HR = 2.94, 95% CI = 1.64–5.57, *p* < 0.001), as were ABOi with DRWRc (HR = 1.87, 95% CI = 1.24–2.82, *p* = 0.003) and ABOi alone (HR = 2.02, 95% CI = 1.42–2.87, *p* < 0.001; Appendix A). The mortality rate due to infection during the study period was higher in the ABOi-DRWRi group than in the other groups, although the rate was low in all groups (0.4%, 0.5%, 1.0%, and 2.4% in the ABOc-DRWRc, ABOc-DRWRi, ABOi-DRWRc, and ABOi-DRWRi groups, respectively, *p* = 0.033).

Although the outcomes of ABOi LDKT have been inconsistent among different studies, patients who undergo ABOi LDKT have better long-term survival than those who wait for an extended period of time for an ABOc transplant from a deceased donor [11]. Furthermore, we recently showed that outcomes of ABOi LDKT were comparable to those of ABOc LDKT and better than those of KT from a deceased donor, even in older patients [22]. These findings suggest that ABOi LDKT should be considered an essential treatment option for patients with ESRD when an ABOc donor is not available. In this context, it is important to identify risk factors that affect the outcome of ABOi LDKT.

ABOi alone was not an independent risk factor for death-censored graft survival in our population, in contrast to the findings of a recent meta-analysis [10]. One reason for this discrepancy might be that overall LDKT outcomes have improved due to the use of tacrolimus-based immunosuppression in the study era. Only 2% of the patients in our study lost their renal allograft during a mean follow-up of approximately 3 years, a much lower rate than in most studies included in the meta-analysis. This excellent outcome was also influenced by our exclusion of HLA incompatible LDKT from the study population, except for XM-DSA+, which was reported to not increase the short-to-medium-term risk of graft loss [23]. Another possible cause for the good outcomes in our study was the relatively high frequency of rituximab use among the patients who underwent ABOi LDKT. In the previous meta-analysis, ABOi LDKT with rituximab-based desensitization resulted in equal graft survival compared to ABOc LDKT. Rituximab was shown to reduce BPAR in immunologically high-risk patients in a prior randomized controlled study [24]. In Korea, rituximab for pre-transplant desensitization is covered by the national health insurance, so most recipients in ABOi LDKTs are treated with rituximab prior to surgery [5,7,25]. Although data regarding rituximab use were available for only 45% of the patients in our study who underwent ABOi LDKT, 94.8% of these patients received rituximab before KT, which might explain the high rate of graft survival.

Among the ABOi LDKT patients in our study, only those with DRWRi had lower death-censored graft survival rates compared with ABOc LDKT patients. Donor-recipient size mismatch is a well-known risk factor in KT [26,27]; however, the risk due to size mismatch alone is low (increased risk by approximately 1.1 to 1.2 times) [16,17] or even insignificant in studies of deceased donor KT in the modern era [28]. Therefore, studies have focused on the combination of size mismatch and other mild-to-moderate risk factors, such as extended donor criteria and sex mismatch. The risk of DRWRi was included in the living kidney donor profile index developed in the United States, but it only reflects an increased risk if the DRWR is <0.9 [29]. In our study, DRWRi was only a marginal risk factor for death-censored graft survival in the multivariable analysis (*p* = 0.074), which could be due to the relatively small population and the short-to-medium duration of follow-up. Regardless, our results showed that DRWRi had a greater effect on graft survival in ABOi LDKT than in ABOc LDKT.

In the Kaplan–Meier analysis, patients without BPAR events during follow-up had equal graft survival regardless of ABOi or DRWRi. In contrast, among patients who experienced BPAR, the combination of ABOi and DRWRi was associated with lower death-censored graft survival, although the difference was not significant, probably because of the small sample size. The low graft survival after ABOi-DRWRi LDKT among patients who experienced BPAR might be due to low functional reserves in relatively small renal allografts, which were well demonstrated in a previous study [18]. More potent rejection, such as antibody mediated rejection due to ABOi, is another possible explanation of the low graft survival after ABOi-DRWRi KT among patients with BPAR. Although our study did not identify different types of acute rejection, our results suggest that the consequences of rejection might be more hazardous in ABOi-DRWRi KT than in KT without both risk factors.

Interestingly, ABOi-DRWRi was the independent risk factor for patient survival while ABOi alone was not a significant risk factor in this study. High recipient BMI, which is characteristic of DRWRi, might affect patient survival; however, DRWRi alone did not significantly increase the risk of mortality. In fact, recent studies in the United States found that recipient BMI was not an independent risk factor for patient survival after KT [30]. A possible explanation for increased risk of patient death in ABOi-DRWRi KT is a higher rate of severe infection due to pre-transplant desensitization and treatment for BPAR. The ABOi-DRWRi group included many cases in which male patients received kidneys from wives with comparatively low BMI and incompatible blood types, so unrelated donors with relatively low levels of HLA match were frequent in that group. Despite an improved understanding of DSA and the development of immunosuppression, the level of HLA matching is still considered a strong risk factor for BPAR and graft survival [31]. In our study, although the difference was not significant in the multivariable analysis, the ABOi-DRWRi group had a higher rate of BPAR than the other groups in the Kaplan–Meier analysis. We hypothesize that immunological treatment for BPAR synergized with pre-transplant desensitization for ABOi and resulted in a higher rate of severe infections. This complication might have been further affected by poor renal function due to DRWRi. ABOi alone was also an independent risk factor for pneumonia, although its HR (2.02) was lower than that of ABOi-DRWRi (2.94). The higher rate of mortality due to infection in the ABOi-DRWRi group than the other groups supports our hypothesis, although this finding should be interpreted cautiously because of the small number of events observed.

Our study had some limitations. Because of the short-to-medium duration of follow-up, our results do not represent the long-term consequences of ABOi and DRWRi. The small sample size of the ABOi-DRWRi group was another limitation, which resulted in large confidence intervals of hazard for some outcomes. Additionally, detailed information, including the strength of anti-blood group antibody and desensitization protocols, was not included in the risk factor analysis because of data unavailability in the registry. Furthermore, potential confounders not controlled by the multivariable analysis because of the small number of events could be present, although we attempted to optimize the adjusted model to obtain statistical relevance.

## 4. Conclusions

The combination of ABOi and DRWRi was an independent risk factor for graft and patient survival after LDKT, whereas neither factor alone significantly affected graft or patient survival. Thus, donor-recipient weight matching should be considered when selecting donors for ABOi LDKT.

## Figures and Tables

**Figure 1 jcm-10-05674-f001:**
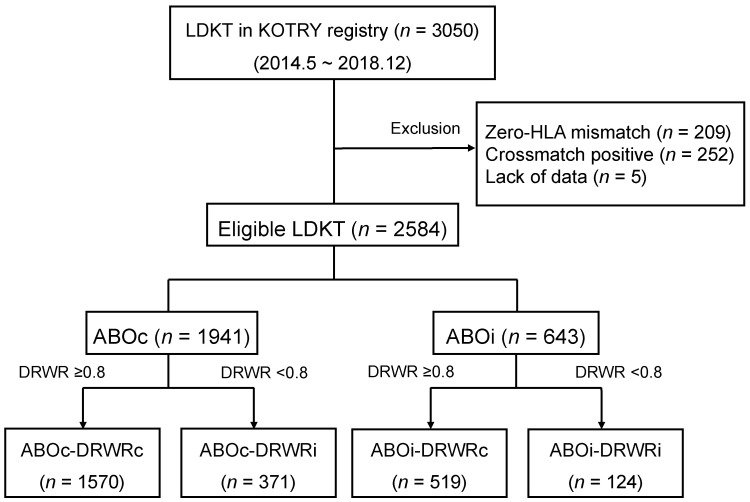
Study population. DRWR, donor-recipient weight ratio; HLA, human leukocyte antigen; LDKT, living donor kidney transplantation.

**Figure 2 jcm-10-05674-f002:**
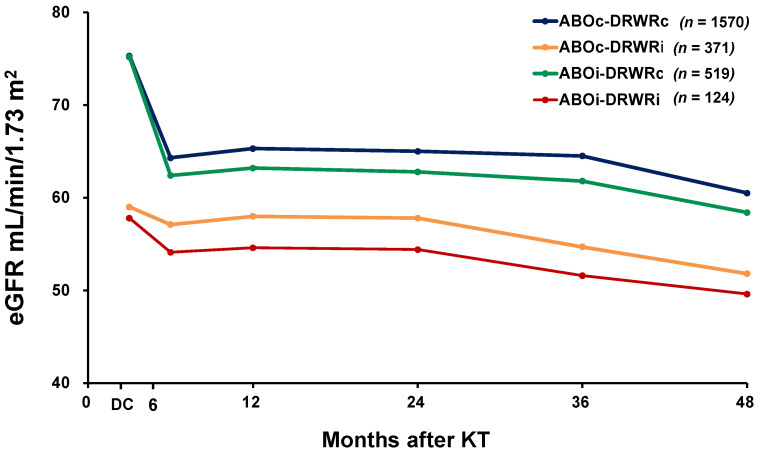
Graft function from hospital discharge to 48 months after KT. *p*-values of ANOVA and pos hoc analyses at each time point are provided in the Appendix A. DC, discharge; KT, kidney transplantation.

**Figure 3 jcm-10-05674-f003:**
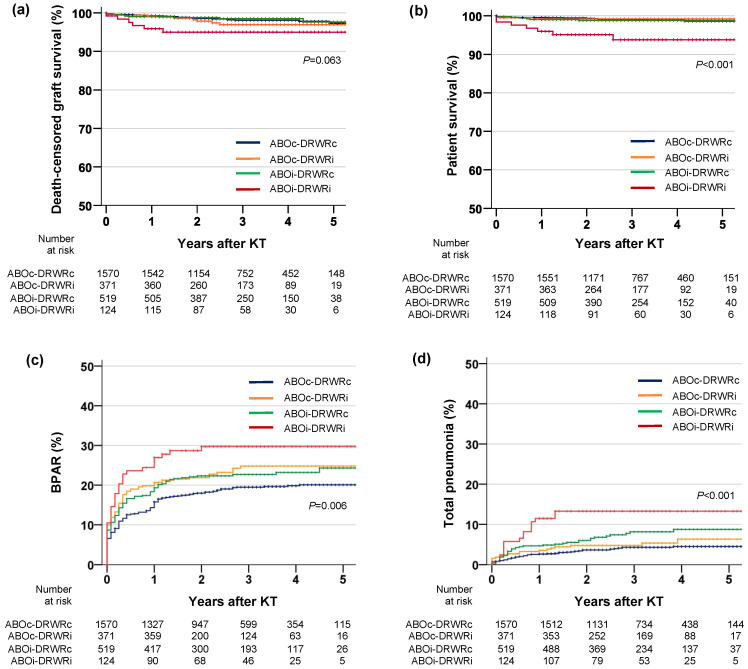
Kaplan–Meier analyses of the outcomes. (**a**) Death-censored graft survival, (**b**) patient survival, (**c**) BPAR, and (**d**) total pneumonia. BPAR, biopsy-proven acute rejection; KT, kidney transplantation.

**Figure 4 jcm-10-05674-f004:**
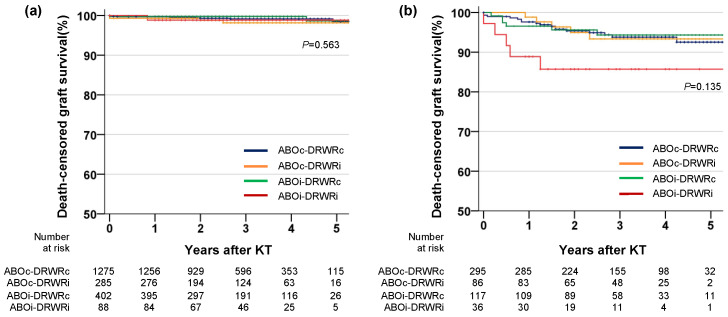
Graft survival among patients who did or did not experience BPAR. (**a**) Did not experienced BPAR, (**b**) experienced BPAR. BPAR, biopsy-proven acute rejection; KT, kidney transplantation.

**Table 1 jcm-10-05674-t001:** Baseline characteristics.

Variables	ABOc-DRWRc (*n* = 1570)	ABOc-DRWRi (*n* = 371)	ABOi-DRWRc (*n* = 519)	ABOi-DRWRi (*n* = 124)	*p*
Age, years	47.6 ± 12.2	47.1 ± 10.7	47.6 ± 12.5	49.8 ± 9.2	0.192
Sex, male	847 (53.9)	345 (93.0)	277 (53.4)	116 (93.5)	<0.001
BMI, kg/m^2^	22.5 ± 3.3	26.6 ± 3.4	22.3 ± 3.3	26.3 ± 3.6	<0.001
Donor age, years	45.6 ± 12.2	48.0 ± 9.9	47.0 ± 12.1	47.6 ± 9.7	0.001
Donor sex, male	804 (51.2)	30 (8.1)	261 (50.3)	8 (6.5)	<0.001
Donor BMI, kg/m^2^	24.7 ± 3.1	22.2 ± 2.4	24.5 ± 3.2	22.1 ± 2.3	<0.001
Recipient-donor relationship					<0.001
Unrelated	596 (38.0)	213 (57.4)	278 (53.6)	98 (79.0)	
Related	974 (62.0)	158 (42.6)	241 (46.4)	26 (21.0)	
Recipient-donor relationship-detail					<0.001
Spouse	487 (31.0)	195 (52.6)	248 (47.8)	91 (73.4)	
Parent	280 (17.8)	61 (16.4)	91 (17.5)	10 (8.1)	
Offspring	345 (22.0)	28 (7.5)	79 (15.2)	7 (5.6)	
Sibling	349 (22.3)	69 (18.6)	71 (13.7)	9 (7.3)	
Other unrelated	109 (6.9)	18 (4.9)	30 (5.8)	7 (5.6)	
Number of HLA mismatch (for A/B/DR)	3.6 ± 1.4	3.9 ± 1.5	3.9 ± 1.5	4.3 ± 1.4	<0.001
Cause of ESRD					0.001
Diabetes	343 (21.8)	120 (32.3)	126 (24.3)	40 (32.3)	
Hypertension	214 (13.6)	51 (13.7)	61 (11.8)	21 (16.9)	
Glomerular disease	568 (36.2)	100 (27.0)	181 (34.9)	33 (26.6)	
PCKD	74 (4.7)	22 (5.9)	30 (5.8)	8 (6.5)	
Other disease	48 (3.1)	4 (1.1)	19 (3.7)	3 (2.4)	
Unknown	323 (20.6)	74 (19.9)	102 (19.7)	19 (15.3)	
Dialysis duration, months	17.8 ± 35.7	14.5 ± 34.5	17.7 ± 36.3	11.5 ± 23.4	0.110
Retransplantation	114 (7.3)	7 (1.9)	28 (5.4)	3 (2.4)	<0.001
Diabetes	441 (28.1)	147 (39.6)	162 (31.2)	53 (42.7)	<0.001
Hypertension	1388 (88.4)	348 (93.8)	467 (90.0)	121 (97.6)	<0.001
CVD	134 (8.5)	48 (12.9)	43 (8.3)	13 (10.5)	0.050
XM-DSA+	84 (5.4)	6 (1.6)	50 (9.6)	8 (6.5)	<0.001
Induction agent					0.330
IL-2 receptor antibody	1390 (88.5)	339 (91.4)	463 (89.2)	107 (86.3)	
Anti-thymocyte globulin	180 (11.5)	32 (8.6)	56 (10.8)	17 (13.7)	

BMI, body mass index; CVD, cardiovascular disease; DSA, donor specific antibody; ESRD, end stage renal disease; HLA, human leukocyte antigen; IL, interleukin; PCKD, polycystic kidney disease; XM, crossmatch.

**Table 2 jcm-10-05674-t002:** Univariable and multivariable Cox analysis.

	Univariable Cox		Multivariable Cox	
Variables	HR (95% CI)	*p*	HR (95% CI)	*p*
**For death-censored graft survival ^a^**				
ABOi vs. ABOc	1.18 (0.64–2.19)	0.601	1.10 (0.59–2.04)	0.770
DRWRi vs. DRWRc	1.87 (1.02–3.43)	0.042	1.75 (0.95–3.23)	0.074
ABOc-DRWRc	Reference		Reference	
ABOc-DRWRi	1.46 (0.69–3.10)	0.327	1.32 (0.62–2.83)	0.469
ABOi-DRWRc	0.90 (0.41–1.98)	0.791	0.82 (0.37–1.81)	0.624
ABOi-DRWRi	2.94 (1.21–7.12)	0.017	2.73 (1.11–6.70)	0.028
**For patient survival ^b^**				
ABOi vs. ABOc	2.18 (1.07–4.45)	0.032	1.65 (0.79–3.42)	0.181
DRWRi vs. DRWRc	2.05 (0.96–4.34)	0.063	1.64 (0.75–3.59)	0.214
ABOc-DRWRc	Reference		Reference	
ABOc-DRWRi	0.86 (0.25–2.97)	0.813	0.67 (0.19–2.35)	0.533
ABOi-DRWRc	1.21 (0.47–3.12)	0.693	0.93 (0.35–2.42)	0.875
ABOi-DRWRi	6.02 (2.46–14.77)	<0.001	3.55 (1.39–9.04)	0.008
**For BPAR ^c^**				
ABOi vs. ABOc	1.24 (1.03–1.50)	0.024	1.14 (0.94–1.38)	0.190
DRWRi vs. DRWRc	1.31 (1.07–1.60)	0.009	1.05 (0.84–1.31)	0.661
ABOc-DRWRc	Reference		Reference	
ABOc-DRWRi	1.29 (1.02–1.64)	0.037	1.02 (0.79–1.32)	0.879
ABOi-DRWRc	1.23 (0.99–1.52)	0.060	1.11 (0.89–1.38)	0.355
ABOi-DRWRi	1.66 (1.17–2.34)	0.004	1.27 (0.88–1.82)	0.203
**For pneumonia ^d^**				
ABOi vs. ABOc	2.08 (1.47–2.93)	<0.001	2.02 (1.42–2.87)	<0.001
DRWRi vs. DRWRc	1.53 (1.04–2.25)	0.030	1.28 (0.85–1.94)	0.235
ABOc-DRWRc	Reference		Reference	
ABOc-DRWRi	1.34 (0.80–2.25)	0.262	1.15 (0.67–1.97)	0.605
ABOi-DRWRc	1.91 (1.27–2.86)	0.002	1.87 (1.24–2.82)	0.003
ABOi-DRWRi	3.53 (2.04–6.12)	<0.001	2.94 (1.64–5.57)	<0.001

Models for each outcome were determined with covariates of which *p*-value was <0.10 in univariable Cox. Full results of Cox analysis were provided in Appendix A. ^a^ Multivariable Cox model for death-censored graft survival included age, donor age, number of HLA mismatches, and XM-DSA+. ^b^ Multivariable Cox model for patient survival was established by stepwise regression for the Optimal statistical significance with limited number of death events (*n* = 30). CVD, unrelated donor and XM-DSA+ were finally included in the model. ^c^ Multivariable Cox model for BPAR included age, sex, retransplantation, CVD, donor age, donor sex, donor BMI, number of HLA mismatches, XM-DSA+, and induction agent. ^d^ Multivariable Cox model for pneumonia included age, sex, CVD, donor age, and unrelated donor. BPAR, biopsy-proven acute rejection; CVD, cardiovascular disease; DSA, donor specific antibody; HLA, human leukocyte antigen; XM, crossmatch.

**Table 3 jcm-10-05674-t003:** Infectious complications within 1 year after kidney transplantation.

Variables	ABOc-DRWRc (*n* = 1570)	ABOc-DRWRi (*n* = 371)	ABOi-DRWRc (*n* = 519)	ABOi-DRWRi (*n* = 124)	*p*
Total infections	321 (20.4)	85 (22.9)	118 (22.7)	30 (24.2)	0.480
Urinary tract infection	152 (9.7)	37 (10.0)	66 (12.7)	9 (7.3)	0.155
Bacterial pneumonia	21 (1.3)	8 (2.2)	10 (1.9)	9 (7.6)	<0.001
Bacteremia	6 (0.4)	4 (1.1)	6 (1.2)	2 (1.6)	0.106
Viral infection	142 (9.0)	37 (10.0)	44 (8.5)	17 (13.7)	0.308
Viral pneumonia	6 (0.4)	2 (0.5)	9 (1.7)	4 (3.2)	0.001
Fungal infection	11 (0.7)	3 (0.8)	7 (1.3)	2 (1.6)	0.451
Pneumocystis jiroveci pneumonia	8 (0.5)	2 (0.5)	7 (1.3)	4 (3.2)	0.005

Using data from a Korean nationwide patient registry, we demonstrated that the combination of DRWRi and ABOi increased the risks of graft loss and patient mortality, whereas neither factor alone increased these risks. Poor graft survival after ABOi-DRWRi LDKT was prominent among patients who experienced BPAR, although the combination of ABOi and DRWRi was not an independent risk factor for BPAR. However, the combination of ABOi and DRWRi was an independent risk factor for pneumonia and infection-related death.

## Data Availability

This national registry data cannot be made available to other researchers according to the policy.

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
