# Peer review of "Risk Due to ABO Incompatibility and Donor-Recipient Weight Mismatch in Living Donor Kidney Transplantation: A National Cohort Study"

_jcm, 2021, doi:10.3390/jcm10235674_

Round 1

Reviewer 1 Report

In this study, Lee et. al. attempted to study the association between donor-recipient weight mismatch focusing on the ABOi recipients. Unfortunately, the sample size specially for those with both incompatibility is very small. As expected the confidence intervals for the hazards are quite large. 

Additionally, while using exploratory approach or doing a uni-variable analysis and including the statistically significant predictors, many known predictors of outcome were not included in the model. 

I suggest considering IPW or other forms of propensity matching to balance the baseline characteristics and to consider including clinically relevant predictors in the final models (graft and recipient survival) e.g. diabetes. 

Also, it was interesting to see the analysis focusing on pneumonia as an outcome of interest and specifically modeling. I suggest to focus on graft and recipient survival. 

Finally, the authors will have to support the chosen cut off of 0.8 for DRWRi. 

Author Response

Response to reviewer #1

In this study, Lee et. al. attempted to study the association between donor-recipient weight mismatch focusing on the ABOi recipients. Unfortunately, the sample size specially for those with both incompatibility is very small. As expected the confidence intervals for the hazards are quite large. 

=> Thank you for your good point of view. Small sample size for patients with both incompatibility was limitation of our analysis. However, we think showing this small KT population has a risk of worse outcome in living donor KT. We added additional limitation in the discussion section, as following next paragraph.

Additionally, while using exploratory approach or doing a uni-variable analysis and including the statistically significant predictors, many known predictors of outcome were not included in the model. I suggest considering IPW or other forms of propensity matching to balance the baseline characteristics and to consider including clinically relevant predictors in the final models (graft and recipient survival) e.g. diabetes. 

=> Thank you for your good suggestions. We also considered IPTW or propensity matching to balance baseline characteristics among study population. However, we found it was inappropriate to perform those statistics with 4 different subgroups. Instead, we selected multivariable Cox regression to control confounders for each outcome. We think this method is statistically relevant either even though baseline factors were not matched among 4 groups.

Furthermore, there were limited number of outcomes, especially graft failure and patient death, in our LDKT population who had extremely good prognosis. Thus, we think including all clinically important factors which were not significant in univariable model into final adjusted model is not statistically relevant. We even did not include variables of which P value was <0.05 in univariable model for patient survival due to small number of events. We already describe the reason for this statistical approach in the method section.

Combined with the reply for prior paragraph, we added next sentence in the discussion section as additional limitation.

<Page 14>

Small sample size of ABOi-DRWRi group and limited number of major events such as graft failure and patient death could weaken the significance of results presented as large confidence intervals of hazard.

Also, it was interesting to see the analysis focusing on pneumonia as an outcome of interest and specifically modeling. I suggest to focus on graft and recipient survival. 

=> Thank you for your suggestion. As your opinion, graft and patient survival are most important outcome in this kind of analyses for KT patients. However, we thought comparing other outcomes such as BPAR and infection would be also valuable for demonstrating the risk of ABOi or DRWRi LDKT. We are sure that such various outcomes are one of strengths of KOTRY data compared to another registry.

Finally, the authors will have to support the chosen cut off of 0.8 for DRWRi. 

=> Because there was no definite cut off for judging small donor for given recipients, we just had to set arbitrary cut off as 25 percentile value among our study population for DRWRi. This is similar way used in prior studies  [1-4].

  1. Arshad A, Hodson J, Chappelow I, Nath J, Sharif A. The Influence of Donor to Recipient Size Matching on Kidney Transplant Outcomes. Transplant Direct 2018; 4: e391.
  2. Miller AJ, Kiberd BA, Alwayn IP, Odutayo A, Tennankore KK. Donor-Recipient Weight and Sex Mismatch and the Risk of Graft Loss in Renal Transplantation. Clin J Am Soc Nephrol 2017; 12: 669.
  3. Goldberg RJ, Smits G, Wiseman AC. Long-term impact of donor-recipient size mismatching in deceased donor kidney transplantation and in expanded criteria donor recipients. Transplantation 2010; 90: 867.
  4. Giral M, Foucher Y, Karam G, et al. Kidney and recipient weight incompatibility reduces long-term graft survival. J Am Soc Nephrol 2010; 21: 1022.

Reviewer 2 Report

Well designed study with big number of patients.

This paper clearly shows that size mismatch it’s a relevant factor in living donor ABOi kidney 

Author Response

There is no comment needing revision from this reviewer.

Reviewer 3 Report

The study is interesting and well presented

Author Response

(The authors gave the same response as above.)

Round 2

Reviewer 1 Report

Authors addressed the raised questions satisfactorily. The expanded limitation section appropriately cautions the reader. Thank you for revising the manuscript.